# Supraglacial debris cover assessment in the Caucasus Mountains, 1986-2000-2014

Levan G. Tielidze<sup>1,5</sup>, Roger D. Wheate<sup>2</sup>, Stanislav S. Kutuzov<sup>3</sup>, Kate Doyle<sup>4</sup>, Ivan I. Lavrentiev<sup>3</sup>

5

<sup>1</sup>Department of Geomorphology, Vakhushti Bagrationi Institute of Geography, Ivane Javakhishvili Tbilisi State University, Tamarashvili st., Tbilisi, Georgia, 0177 <sup>2</sup>Natural Resources and Environmental Studies, University of Northern British Columbia, 3333 University Way, Prince George, BC, Canada, V2N 4Z9

<sup>3</sup>Department of Glaciology, Institute of Geography of the Russian Academy of Sciences, 29 Staromonetniy Pereulok, Moscow, Russia 119017

<sup>4</sup>Institute of Cartography, Technical University of Dresden, 01069, Dresden, Germany
<sup>5</sup>Department of Earth Sciences, Georgian National Academy of Sciences, 52 Rustaveli Ave., Tbilisi, Georgia, 0108

15

Correspondence to: Levan G. Tielidze (levan.tielidze@tsu.ge)

# Abstract

- Surpaglacial debris cover plays an increasingly important role impacting on glacier ablation, while there have been limited recent studies for the assessment of debris covered glaciers in the Greater Caucasus mountains. We selected 559 glaciers according to the sections and macroslopes in the Greater Caucasus main watershed range and the Elbrus massif to assess supraglacial debris cover (SDC) for the years 1986, 2000 and 2014. Landsat (Landsat 5 TM, Landsat 7 ETM+, Landsat 8 OLI) and SPOT satellite imagery were analysed to generate glacier outlines using
- <sup>25</sup> manual and semi-automated methods, along with slope information from a Digital Elevation Model. The study shows there is greater SDC area on the northern than the southern macroslope, and more in the eastern section than the western and central. In 1986-2000-2014, the SDC area increased from 6.4%-8.2%-19.4% on the northern macroslope (apart from the eastern Greater Caucasus section), while on the southern macroslope, SDC increased from 4.0%-4.9%-9.2%.
- 30 Overall, debris covered glacier numbers increased from 122-143-172 (1986-2000-2014) for 559 selected glaciers. Despite the total glacier area decrease, the SDC glacier area and numbers increased as a function of slope inclination, aspect, glacier morphological type, Little Ice Age (LIA) moraines, rock structure and elevation. The datasets are available for public download at doi.pangaea.de/10.1594/PANGAEA.880147.

### **1** Introduction

Mountain glaciers are an integral part of the cryosphere and a vital component of Earth's natural systems, serving as sensitive indicators of climate change. Glaciers around the world have been experiencing recession at varying intensities and offer explicit evidence of global warming

5 (Haeberli, et al., 1999; Oerlemans, 2005; Paul et al., 2007). Due to their remoteness, size and inaccessible nature, remotely sensed data are likely the most effective tool for regular mapping of mountain glaciers in a comprehensive manner.

Glacier surfaces can be either clean or debris-covered, where the sources and quantity of debris are variable. The prime activities which cause debris cover on glaciers are rock fall and
slides, and mass movements from adjacent mountain slopes, with other sources including pollutants, salts and micro-organisms from sea spray, volcanic eruptions and wind-blown dust (Benn and Evans, 1998). Snow and rock-ice avalanching are also an important mechanism for transporting debris onto glacier surfaces (Raina and Srivastava, 2008), especially in high mountain environments such as the Greater Caucasus, where large quantities of snow can accumulate on unstable slopes (Kaldani and Salukbadze, 2015). When these snow or rock-ice slopes collapse, they can result in large and destructive avalanches, enough to detach debris from underlying rock surfaces, e.g. the Devdoraki Glacier rock-ice avalanche in Georgia in May 2014 (Tielidze, 2017a).

Generally, debris covered glaciers characterize most mountain regions of the world (Scherler
 et al., 2011) and are an inseparable component of glacial systems in mountain environments (Biddle, 2015). Supraglacial debris cover (SDC) obscures glacier ice extents, challenging accurate glacier mapping (Shukla et al., 2010a; Veettil, 2012). Additionally, debris cover on the tongues of mountain glaciers affects melt rates, increasing rates of ablation in cases of thin debris cover, or decreasing ablation under thick debris cover (Brock et al. 2010). Changes in glacial environments

in mountain regions, especially with regard to debris cover, can augment the potential for glacial hazards (Benn et al., 2012). For some regions where the local population is dependent on glacial meltwater for water supplies, exact evaluation of glacial hydrology is important to ensure the sustainable use of water resources (Baraer et al., 2012). The difficulty of such investigations is associated with poor knowledge of the large-scale spatial distribution of the thickness and properties of debris, since field measurements of thickness and properties of the debris layer have practical difficulties on a large scale, and methods for satellite mapping of supraglacial debris remain in development (Foster et al., 2012; Zhang et al., 2016).

Europe's highest mountain system - the Greater Caucasus - contains over 2000 glaciers, with a total area of 1121±30 km<sup>2</sup> (Kutuzov et al., 2015). The Greater Caucasus SDC is an important control for ice ablation, as it is similarly in many other glaciated areas (Lambrecht et al., 2011) and has been identified as a key player in glacier mass balance (Popovnin and Rozova, 2002). In addition, in some cases SDC and proglacial lakes are directly related to glacial hazards. Therefore, it is necessary to take into account the SDC and periglacial debris cover on the outside

of the glacier margin, when assessing temporal change in this region. Debris cover becomes especially important in understanding the complex relation between climate change and glacier accumulation and ablation.

#### 2 Study area and previous studies 5

The Greater Caucasus can be divided into three parts: Western, Central, and Eastern. Their borders run near the meridians of Mount Elbrus (5642 m) and Mount Kazbegi (5047 m) (Tielidze, 2017b). At the same time, the terms Northern and Southern Caucasus are frequently used to refer to the corresponding macroslopes of the Greater Caucasus range (Solomina et al., 2016).

10 As a result of variations in climate conditions, glacier character and rock structure (Tielidze, 2017a), both northern and southern macroslopes were selected for the western and central Greater Caucasus, as they contain most of the glaciated areas. However, for the eastern section, only the northern macroslope was selected as glaciers are almost non-existent in the south (Fig. 1).

In the western Greater Caucasus, 145 glaciers in the Kuban River basin (northern slope) and 78 glaciers in Kodori River basin (southern slope) were selected (Fig. 1a); 173 glaciers in the 15 Baksan, Chegem, Cherek (northern slope) and 112 glaciers in the Enguri River basin (southern slope) were selected for the central Greater Caucasus (Fig. 1b); and 130 glaciers in the Tergi (Terek) headwaters, Sunja Right tributaries and Sulak river basins were selected in the eastern Greater Caucasus (Fig. 1c). In addition, 21 glaciers on the Elbrus massif were selected as an area of less debris cover (Fig. 1d). The size of the largest glacier selected was 37.5 km<sup>2</sup> and the

20

smallest 0.01 km<sup>2</sup>.

Figure 1. Investigated area (a) and selected glaciers in sections – western Greater Caucasus (b); central Greater Caucasus (c); eastern Greater Caucasus (d); the Elbrus (e). 25