# Peer review of "Supraglacial debris cover assessment in the Caucasus Mountains, 1986-2000-2014"

_Earth System Science Data, 2017_

## Short Comment (SC1) · 20 Sep 2017

This research detects the glacier (supraglacial debris cover also) change in the Greater Caucasus mountains region based on Landsat images. This work is significant because inventorying of glacier (supraglacial and clean ice) area and detects its change can help us understand the implications of continued and exacerbated climate change. However, there have several comments on the methodology part. There are several methods (selected based on the previous studies) were used to map the glacier in the study area. This research tries to find the best suitable method for accurate detects the supraglacial debris area. Such as Thermal/Near-IR/Mid-IR band ratio methodology (Alifu et al. 2015), semi-automated classification methodology using geomorphometric parameters and Landsat 8, and, manual delineation compared to

Red/Mid-IR ratio methodology were tested. However, the application of the method proposed by Alifu et al. 2015 was incorrect. Frist, (Alifu et al. (2015) devised a compound ratio method to reduce these errors, dividing digital number (DN) values of the thermal band (band 6) by the standard red/Mid-Infrared ratio, for 15 Landsat TM and ETM+ [Ratio = Band 6 / (Band 4 / Band 5)], and applying a threshold value of 2.0.) Page5, Line 15. Alifu et al. 2015 used DN values in the new band ratio image from 137 to 180 (Figures 4(b) and 6(b)) and 140 to 234 (Figures 5(b) and 7(b)) were used as the thresholds for mapping the supraglacial debris areas in the Koxkar glacier and Yengisogat glaciers. Second, Thermal/Near-IR/Mid-IR band ratio alone cannot accurately detect the supraglacial debris area. Therefore, additional information was needed such as combination with geomorphometric parameters (Alifu et al. 2015, Alifu et al. 2016), then, manual editing is required. Third, the improper threshold value used to map supraglacial debris area (example, page9, figure 4). Also, selected bands for generated the Thermal/Near-IR/Mid-IR band ratio using Landsat 8 images is b10/b5/b6 (https://landsat.gsfc.nasa.gov/landsat-data-continuity-mission/). Figure 1 (left) is same area with figure 4.a (page9). Figure 2 (right) is same area with figure 4.d (page9).Please see figures in the Supplement. Green colored area in figure 1 and the blue colored area in figure 2 are supraglacial debris area. However, the result of the density slice contained several inaccuracies for the 'supraglacial debris' class in areas where bedrock valley walls were located in shade and/or in higher elevation areas. These classification errors could be eliminated when slope information was also considered. Selecting the threshold values is a critical step for delineating the debris-covered glacier accurately, but the threshold values were shown to differ from glacier to the glacier and Landsat images. Therefore, the thresholds should be selected carefully by overlaying the density-sliced maps with Landsat composite images and other ancillary data. Forth, although, the method proposed by Alifu et al. 2015 mapping of debris-covered glaciers with promising accuracy, however, the combination of Thermal/Near-IR/Mid-IR band ratio and geomorphometric parameters have limitations, larger inaccuracy occurred in the small debris-covered glacier. Finally,

this is a valuable contribution and with the changes made it will be useful to the glacial community. Also, I do feel your article can contribute greatly to current glacier research and is worth publication.

Please also note the supplement to this comment:
https://www.earth-syst-sci-data-discuss.net/essd-2017-96/essd-2017-96-SC1-supplement.pdf

---

## Referee Comment (RC1) · Anonymous Referee #1 · 5 Oct 2017

General comments: A thorough language editing is needed as there are issues and awkwardness in the sentence structure right from the first sentence of the abstract. I will not be pointing out them separately as they are present in almost every other sentence. There are several issues with the methods too as highlighted by Alifu, H. in his interactive comments. In the light of these issues, presently it is really difficult to comment upon the accuracy of the results. Also, how is this a review article as I can see in the manuscript type? It certainly is a research article and not a comprehensive review.

Specific comments: Below I provide some specific recommendations and queries: Abstract The article is about debris cover and therefore it is important to state its relevance properly instead of mentioning it in sweeping terms in both, abstract as well as intro-

duction. The debris cover is relevant to study not just because of its impact on the glacier ablation but also because it is considered to be a significant part of an efficient sediment transport system (supraglacial, englacial, and subglacial) in cold and high mountains which ultimately affect the overall dynamics, and mass and energy balances of the glacier. Several studies have also reported debris cover's role in promoting the formation of supraglacial lakes. These mentions should come in the abstract and should be more detailed in the introduction along with proper references. Full forms of TM, ETM, SPOT, OLI, ALOS, ASTER etc. are not provided anywhere.

Introduction It is presently fragmented with small and incomplete paragraphs. Many relevant and recent references on debris cover mapping have been overlooked. The last paragraph of the introduction needs to talk about the present study, how is it filling the research gaps, and what is the structure of the paper. Study area Figure 1: The caption is very oddly structured. Usually the (a), (b), (c)... come before their descriptions. Landsat 8 is the correct name and not Landsat L8. P4L6: Spot must be in all capitals.

Data and methods P4L25-29: "...2016 (SPOT) for manual correction of glacier outlines..." which year's outlines? How does using 2016 image to correct the boundaries delineated on past images make sense when 2016 is not even a part of the temporal analysis? Also what was the footprint of the SPOT scene? It needs to be shown in Fig. 1. I am sure that it must have covered very few glaciers than the number that this study is trying to achieve. In that case, modifying the outlines of just a fraction of the total glaciers studied is only going to bring in more inconsistency in the analyses. Paul et al. (2013) statement that "high resolution imagery and manual delineation do not necessarily provide better accuracy, but they certainly provide a better understanding of the difficulties in mapping debris covered glaciers" is a very generalized statement. Glacier mapping results certainly get improved if the manual delineation is performed using high resolution images and of course a lot depends on the interpreter. No matter how much advocacy is presented in favor of semi-automated methods on Landsat or

ASTER images, manual delineations on high resolution images still provide the best results. Certainly an approach using semi-automated followed by manual modifications is faster and preferable. Particularly, if you want to justify your use of SPOT image, then you should refrain from adding this Paul et al. (2013) sentence here as it is not only an overly sweeping statement but is also contradicting your actions. Why was the Ground Penetrating Radar (GPR) data used and how was it relevant? Also, what was the coverage of the GPR survey and it needs to be marked in fig. 1. P4L30-35: I agree with your choice of ASTER GDEM V2 over SRTM. Please add some references such as "Kääb, A., Treichler, D., Nuth, C., & Berthier, E. (2015). Brief Communication: Contending estimates of 2003–2008 glacier mass balance over the Pamir–Karakoram–Himalaya. The Cryosphere, 9(2), 557-564. (https://www.the-cryosphere.net/9/557/2015/tc-9-557-2015.html)" which talk about the penetration issues with SRTM DEM. Also provide reference foe the DEM accuracy that you have mentioned. Table 1: Are the authors sure that 28/07/2000 scene was free of seasonal snow? I doubt it and I would like to see a screenshot of it with respect to August or September scene of the same year. Section 3.2: Many recent and relevant papers are missing here apart from a flawed description of Alifu et al. (2015) method. "...and three different combinations were attempted for OLI imagery 20 (23/08/2013) using either of the two thermal bands as well as an average of both bands: on Landsat 8..." What was the result of this exercise and finally which option was used? In any case, spectrally speaking, taking the average of two thermal bands does not make any sense. P6L15-19: I am really confused here. For what exactly the ASTER GDEM was used and why am I again see the mention of another DEM (ALOS) here? Why was there a need to use 2 different DEMs? What is the reported accuracy of ALOS DEM and why should we not suspect that it will also show some snow or ice penetration? Figure 2 is highly confusing. What does the use of Bands 5 and 8 mean here when they are nowhere described within the text?! Removing water bodies and shadows using single bands and not the indices is very unusual and highly doubt the results. P7L1-10 and section 3.4 are actually a part of the discussion and have been wrongly placed here. P9L4-11: Description of the validity

and use of both, the GPR and SPOT data are vague and unclear.

Results Figure 12: Who did the GPR survey? How was the data processed? Where is the microwave profile? It really seems unconvincing seeing a depth of 220 m in the ablation zone of a mountain glacier!

As I mentioned here, presently the used methods are too vague and confusing to really assess the results and the discussion. I agree that the debris cover has increased but then it is happening almost everywhere. Also, the present conclusion is way too generalized! Of course we need monitoring using high resolution images for all the glaciers and it is not a conclusion which is specific for the study area here. Authors have failed in describing the region-specific needs for performing such a study and its long term implications for the communities.

---

## Referee Comment (RC2) · Anonymous Referee #2 · 12 Dec 2017

This paper focuses on assessing multi-temporal trends in area supra glacial debris cover (SDC) for the years 1986, 2000 and 2014 using a variety of imagery (Landsat series, SPOT). They also track the change in the number of SDC glaciers, and attempt to give explanations for the differences in behavior from the northern to southern slopes. Given that changes in the debris cover in this area are not very well documented, this is a noteworthy effort. However in spite of the efforts undertaken to process the Landsat scenes and to try to explain the topographic controls, the paper does not feel very focused; the information could be selected in order to answer scientific questions and perhaps remove information that does not feel relevant.

While the focus of the paper seems to be on the SDC changes, much of the paper discusses differences between various techniques of mapping SDC, which seems to

be a different topic altogether and distracts from the main objective. I suggest that instead of presenting the various approaches, which do not bring much innovation, the authors just choose their approach and conduct an uncertainty analysis and then focus on the area change.

The main concerns are: - Objectives need to be clarified, and the methods and results adjusted accordingly - The excessive increase in SDC seems questionable, particularly with respect to changes in the internal rock area and overall the methodology used - Regional trends reported here so not yield detail about glacier-by-glacier area changes which might be subject to SDC mapping errors - GPR section on ice thickness seems irrelevant here and is outside the scope of the paper, it does not fit with the rest of the analysis - A comparison with other studies focused on area changes is missing, and it is not clear if this study is in line with other studies regarding the increase in area changes

Specific comments are below:

Abstract

The abstract is quite clear but then the paper diverges a bit from it. The authors may specify why they chose to focus on different parts of the ranges, to compare and contrast. The number of debris-covered glaciers is really not important, as there are many processes that could be involved ie. glaciers might disintegrate rather than form and grow within the period of a few decades.

Introduction

The introduction lacks focus, despite bringing in some background information about the importance of glaciers. For ex. the issue of debris thickness is mentioned but the objectives of the paper do not target debris cover thickness. The introduction would need to be revised/focused. Most of the statements need to be expanded since they do not fully explain the gaps in knowedge, and remain vague, for example p 2 lines 31

to 37:

"The Greater Caucasus SDC is an important control for ice ablation, as it is similarly in many other glaciated areas (Lambrecht et al., 2011) and has been identified as a key player in glacier mass balance (Popovnin and Rozova, 2002)."

This would need to be expanded and the concepts explained.

Similarly: "In addition, in some cases SDC and proglacial lakes are directly related to glacial hazards" This remains vague and issues are mentioned but not developed.

Study area and previous studies

The different parts of the range are mentioned but it is not clear how these were defined. The 1st paragraph on p.4 is indented to be a literature review but it mixes glacier area change with debris cover mapping, these are not clearly addressed. The authors could list the studies that determined area changes, the extent of the areas and the remaining gaps; then the debris cover mapping issues could be addressed separately.

P4 l.12- 15 "This research further aims to compare glacier mapping using manual and semi-automated methods to assess the SDC change across a larger area than previous studies according to the western, central and eastern Greater Caucasus over the last 30 years."

This paragraph sounds like it belongs to introduction, not study area/previous studies.

Also, here the objectives do not come out clearly; while in the abstract and introduction the author mention the need to document glacier area changes, here this points to another objective, which is to compare various methods to delineate debris cover.

Overall I suggest that the authors sharpen their objective and focus on glacier area changes in the various parts of the ranges, perhaps splitting the glaciers into clean and debris cover glaciers. To achieve this objective, the authors could just choose their best method for delineating debris cover and provide an uncertainty analysis. As

written currently, this almost contains two different studies and it prevents this from developing either.

3. Data and methods

Sections 3.2 to 3.4 summarize various methods for delineating debris cover, and it seems to be a literature review. This is fine for a paper that would focus on comparing various methods, but here the objective seems to be different and this much detail might not be needed. Sections 3.2 to 3.4 could be combined and shortened, and the authors could just present the approach they took in a section on glacier clean ice and debris cover delineation, for example.

Furthermore: the method tested is presented for the 2013 image, and it's not clear if the same method was applied also to the other images. These sections can be clarified.

P5 l 13- 16 : I do not see the use of presenting the Alifu et al 2015 methodology here, if it is shown later (in results) that their method did not work for this study. I suggest removing the comparison with the Alifu paper in this section as well as in results (p.9 l .16-19). Again, this distracts the reader from the main goal of the paper which gets buried into the details of SDC mapping.

Section 3.5 on comparison with manual digitization can also be presented as part of the glacier mapping section.

Fig.3 seems to belong to the results section, or can be presented as part of the uncertainty analysis.

The SPOT imagery used to ]check the SDC tongues is from 2016, but the Landsat imagery was from 2014- are the authors certain that there was no change in the glacier extents? Area changes might be small but I am not sure this can be used as a check for the classification if the images are not from the same year..

4. Results and discussion

section 4.1 Supraglacial debris cover (SDC) assessment using Thermal/Near-IR/Mid-IR band ratio methodology

Again the focus here is not on assessing the accuracy of the delineation methodology but on the changes in debris cover, it seems- in this case it is unnecessary to mention the method used in the section title.

p. 9 l 16- 25: see my comments above about Alifu et al paper. I suggest removing this comparison and just focusing on the method presented, it the authors are confident in it.

Fig. 4 in the manual outlines, it is not clear if the NW part of the glacier tongue delineated as debris cover is supraglacier debris or just moraine. Have the authors checked this area with the high-resolution imagery? I suggest that in fig 4 it would be more helpful to present the high-resolution imagery rather than the Alifu et al outlines.

P10 l 4-5 "Therefore, the Alifu et al. methodology to identify debris covered ice in the Greater Caucasus cannot be 5 considered robust for OLI imagery, and unsuitable for extended time series". Again the evaluation of Alifu et al method is beyond the scope of the paper as I understand it, and the authors would benefit from focusing on their own method here.

P10 section 4.2 and 4.3 similar to my comments about the methodology, the paper wonders about here with the comparison of the different methods. I think one method should be chosen over the other and the results of that classification method should be chosen.

Table 2: - total glacier number should be "total number of glaciers" - glacier number should be "number of glaciers" since the authors do not refer to glacier IDs - "debris cover should be "debris covered ice"

Figure 6 does not tell much in its present form, sicne there is almost no change. This could be just mentioned in the text, with the

Results on SDC p. 11-12 are hard to follow, and in some parts are descriptive, some ways of improving include: - Separating the glaciers only by range rather that range and glacier area. For ex, in figures 7 and 8 it is hard to distinguish any difference among the rages since different colors are used for the same class (DC) - Clarifying what "percent" the authors are referring to- sometimes this relates to - It would help if instead of average slope, the authors did a correlation between slope of each glacier and its - The large changes (+49

Fig 11 a: The multi-temporal glacier mapping extents shown here are questionable, looking at the area covered by the nutatak. It seems that a) either a lot of the nuatak was exposed in 2016 by surface lowering, or b) that the 1986 image has a lot of snow which covered some of the area around the nutatak. In this case, the area changes are affected by the way that the nunataks were mapped.

For consistency, a number of authors chose to consider the area of the nunataks constant and only estimate the area changes due to glacier retreat. This issue should be carefully assessed since it can be introduce large errors in the change estimates.

Fig 12: the glacier outlines overlaid on the SPOT imagery do not seem to much; it is hard to see but it seems like the SDC area is over-estimated (unless there was a large area change sine 2014 which is unlikely).

Also I am not sure how the GPR analysis helps here, it seems to have been added to the paper since the measurements were taken; however the purpose of the paper is not to provide ice thickness measurements; The GPR section seems to be out of the purpose of the paper and can be removed/saved for a future paper on glacier thickness.

Fig 13: looking at the areas of increase in DC, it seems that some of the "apparent" increase is indeed due to difference on mapping. For ex fig 13b shows an increase in DC area in the glacier accumulation area, which seems to be rather due to the way that the rock outcrops were mapped. Similarly there is a large increased where two glacier tongues marge- this can be a similar case of mapping differences.

Conclusions :

yr +1 should be yr -1

Given the arguments above, I question the main conclusion about the significant increase in SDC in the Caucasus- I think an uncertainty analysis is pertinent before conclusions, and a careful assessment of mapping consistency should be addressed. Furthermore: have any other studies reported an increase in SDC in this area? This should be included in the discussion of the paper to asses whether this is a trend noticed in other studies as well.

---

## Editor Comment (EC1) · RD Drews (Editor) · 13 Dec 2017

Dear authors,

First of all I would like to apologize for comparatively long time of the refereeing/discussion phase. This was because one reviewer had understandable reasons for a delayed report, and I thank the authors for their patience.

Both referee reports and the interactive comment have pointed out significant problems in the methodology (and overall style of the manuscript), making it difficult to assess the principal results of the study. Although some of these concerns may be addresses in a substantial revision of the current paper, I am reluctant to advise you to do this in the framework of ESSD. ESSD's focus is the publication of comprehensive, high-

quality datasets for further reuse in Earth system sciences, and any ambiguities in the methodology (or different interpretations thereof) are a major obstacle for peer-reviews publication. From my point of view, the criticism that was raised can be much better dealt with in a different journal that does not have this data focus (and hence offers more room for data interpretation linked to the applied methodology).

My remarks mentioned here do not preclude that you reply to the comments in full length and provide a revised version. If you do so, the paper will go to re-review. I only wanted to give you an early impression from my side, given that we already lost some time in a lengthy review process. I am sorry that I cannot be more positive at this stage.

Kind regards, Reinhard Drews

———————————————